# New Insights into Mechanisms of Long-term Protective Anti-tumor Immunity Induced by Cancer Vaccines Modified by Virus Infection

**DOI:** 10.3390/biomedicines8030055

**Published:** 2020-03-06

**Authors:** Volker Schirrmacher

**Affiliations:** Immune-Oncological Center Cologne (IOZK), D-50674 Cologne, Germany; V.Schirrmacher@web.de

**Keywords:** antigen-presenting cell, CTL, memory T cell, bone marrow, dendritic cell, oncolytic virus, immunogenic cell death, T cell costimulation, NDV, type I interferon

## Abstract

The topic is how to achieve long-term protective anti-tumor immunity by anti-cancer vaccination and what are its mechanisms. Cancer vaccines should instruct the immune system regarding relevant cancer targets and contain signals for innate immunity activation. Of central importance is T-cell mediated immunity and thus a detailed understanding of cognate interactions between tumor antigen (TA)-specific T cells and TA-presenting dendritic cells. Microbes and their associated molecular patterns initiate early inflammatory defense reactions that can contribute to the activation of antigen-presenting cells (APCs) and to costimulation of T cells. The concommitant stimulation of naive TA-specific CD4+ and CD8+ T cells with TAs and costimulatory signals occurs in T-APC clusters that generate effectors, such as cytotoxic T lymphocytes and T cell mediated immunological memory. Information about how such memory can be maintained over long times is updated. The role that the bone marrow with its specialized niches plays for the survival of memory T cells is emphasized. Examples are presented that demonstrate long-term protective anti-tumor immunity can be achieved by post-operative vaccination with autologous cancer vaccines that are modified by virus infection.

## 1. Introduction

Like the brain with its network of neurons, the immune system has the capacity to learn and develop memory. Both phenomena are prerequisites for effective vaccination and long-term protective immunity. Immunity is systemic, because it is based on the ability of lymphocytes and other immune cells to circulate among tissues. A vaccine that is administered to the skin or muscle can protect from infections in any tissue.

T cell mediated immunity is of relevance for anti-tumor immunity. There are three participants in the molecular recognition of antigen by T cells: an antigenic fragment (peptide) that forms a complex with a presenter molecule (major histocompatibility complex (MHC) protein), and this complex is recognized by a recognition molecule, the antigen-specific T cell receptor (TCR) [1]. During the maturation of T cells in the thymus, negative and positive selection mechanisms lead to central tolerance and ensure that only those mature T cells leave this organ whose TCRs recognize self MHC molecules in association with non-self peptides [2]. Out of the large number of self MHC molecules that exist on cells of an individual, a single T cell recognizes only one, which displays the specific peptide (i.e., MHC restriction of T cells).

The strategies for development of cancer vaccines are partially based on knowledge that was obtained during the development of vaccines to fight infectious diseases. Prophylactic vaccination has been the most effective means of controlling infectious diseases, such as measles, mumps, diphtheria, or polio. Microbial molecular structures are often incorporated into cancer vaccines to initiate fast innate immune reactivity which can help a tumor targeted adaptive immune response.

This review deals with cancer vaccines that are modified by infection with a natural attenuated oncolytic virus (OV). Nevertheless, this relatively simple procedure requires high quality (GMP) production, but it does not involve genetic engineering techniques. The avian Newcastle disease virus (NDV) is a prototype OV that was used to develop cancer vaccines modified by virus infection. Virus-based cancer vaccines involving genetic engineering [3,4] is a different concept and it will not be discussed here.

## 2. Strategy of Designing a Tumor Vaccine Modified by Virus Infection

The objective is to stimulate T cell mediated immune responses against TAs, in particular against tumor neoantigens [5]. Such antigens are peptides that are associated with MHC molecules (pMHC). They are derived from internal proteins of tumor cells. The immunogenicity of proteins requires the recognition of more than one antigenic determinant, as discovered 50 years ago [6]. It is likely that this also holds true for the immunogenicity of TAs and tumor cells. One antigenic determinant, a hapten or a CD8+ T cell epitope is recognized by a B cell receptor (BCR) or CD8+ T cell receptor (TCR), respectively, while the other, an MHC class II associated peptide, is recognized by a CD4+ TCR. Cognate cell-cell interactions between a CD4+ T helper (Th) cell and an antigen-presenting B cell or between an antigen-presenting dendritic cell (DC), a CD4+ Th cell and a CD8+ T cell, then leads to a successful humoral or cell-mediated immune response, respectively.

The strategy of designing a tumor vaccine that is modified by virus infection is based on the two-signal theory of immune activation [7]. T cell responses against TAs require help in the form of costimulation, type I interferons and cytokines to avoid the induction of T cell tolerance. The aim is to induce a strong T helper 1 (Th1) polarized T cell response [8].

A native oncolytic virus (OV) from birds, NDV, an RNA paramyxovirus, is used for tumor cell infection, because it has immunostimulatory properties and it is not adapted to the human immune system [9,10].

Innate immunity is directed towards molecules that are shared by groups of related microbes (pathogen-associated molecular patterns (PAMPs)) and to molecules that are produced by damaged host cells (damage-associated molecular patterns (DAMPs). Such molecules are recognized by germ-line encoded pattern recognition receptors. Tumor cell infection by NDV introduces foreign viral 5′-phosphorylated RNA into its cytoplasm. Such viral RNA is recognized by cytosolic retinoic acid-inducible gene I (RIG-I) receptor [11,12]. The innate antiviral immune response that is initiated by RIG-I activation serves to program a specific adaptive immune response against RNA viruses. Such a physiological program can be exploited to facilitate adaptive immune responses against tumors.

Two compartments of a cell with different types of MHC molecules distinguish the mode of antigen processing and presentation: The cytoplasm and the vesicular compartment (Table 1). This is of relevance for effective anti-tumor immune responses, because these require the activation of two types of T cells: CD8+ cytotoxic T lymphocytes (CTLs) and CD4+ Th cells. Class I MHC molecules present small peptides (e.g., nonamers) from cytosolic proteins, while class II MHC molecules display longer peptides that are generated in vesicles [1]. 

DCs play an important role in the defense against microbes. Table 1 lists three subtypes of human DCs, their characteristic surface markers, transcription factors [13], and their typically released cytokines. They exert antigen-presentation function [14,15] for T cells in the fight against viruses (in particular plasmacytoid DCs), bacteria (conventional DCs), or against virus-infected cells, which could also be virus-infected tumor cells (inflammatory DCs). The classical routes of antigen processing and presentation by DCs are those of class I MHC (e.g., virus infection) in the cell’s cytoplasm and those of class II MHC (e.g., bacteria) following uptake into endosomal/lysosomal vesicles. Inflammatory DCs (iDCs) can ingest virus-infected cells and present antigens via both pathways to CD4+ and CD8+ T cells. iDCs cross-present antigens from the cytosol via the class I MHC pathway. Antigen cross-presentation is of particular relevance for the two types of virus-modified cancer vaccine, whose application induces long-term protective anti-tumor immunity, and which will be described in the following section. 

Cognate interaction of iDC APCs with CD4+ T cells leads to antigen-specific T cell activation and DC1 and Th1 polarization. Three major subsets of CD4+ effector T cells function in host defense against distinct types of infectious pathogens. The development of Th1 cells starts with IL-12 and IFN-γ, cytokines, which are produced by DCs, macrophages, and NK cells. The interaction of these cytokines with respective receptors on naïve CD4+ T cells leads to the activation of the transcription factors T-bet, STAT1, and STAT4, which stimulate differentiation towards Th1. IFN-γ that is produced by the Th1 cells amplifies this response and inhibits the development of Th2 and Th17 cells [16]. Th1 CD4+ T cells interact with CD8+ T cells and help their differentiation into CTLs and CD8+ memory T cells (MTCs) [17]. This T-T cell cooperation leads to the amplification of the polarized cell mediated immune response.

Costimulatory receptors on T cells, like CD28 (receptor for CD80 and CD86) and the TNFRSF9 receptor CD137, are the regulators of metabolism [18]. They can modulate glycolysis, mitochondrial respiratory capacities, and fatty acid β-oxidation, all of which enhance antitumor performance [19]. CD28 engagement induces signaling pathways that enhance TCR signals, leading to the upregulation of cell survival proteins, secretion of IL-2, and expression of its receptor IL-2R (CD25), thus promoting cell proliferation and inducing differentiation to effector cells and MTCs [20]. 

An autologous tumor cell vaccine, as modified by infection with the avian paramyxovirus NDV, termed ATV-NDV, has been developed in animal models [21] and later transferred into the clinic [22]. Additionally, a DC based vaccine, termed IO-VAC^R^, has been developed. The DCs are pulsed with patient-derived tumor cell lysate that was obtained by infection with an oncolytic strain of NDV (i.e., an oncolysate) [23]. It is an approved advanced therapy medicinal product (ATMP), allowed to be administered to human by IOZK, Cologne, on a compassionate use basis. 

## 3. Target Structures in the Virus-modified Vaccines ATV-NDV and IO-VAC^R^

### 3.1. Peptides from Tumor Neoantigens

The majority of tumor antigens (TAs) that elicit protective immune responses are neoantigens that are produced by mutated genes in different tumor cell clones. The term neoantigen [5] means that the antigenic epitope of the TA has not been previously affected by central or peripheral tolerance mechanisms within the host’s immune system. Peptides from neoantigens represent non-self peptides and they are presented as unique pMHC complexes.

The virus-modified vaccines present TAs from autologous tumor cells that are based on the findings that protective anti-tumor immunity induced by such vaccines is specific for autologous tumor cells [24].

### 3.2. Viral PAMPs 

NDV introduces the following PAMPs into the vaccine: foreign viral RNA [11,12] and viral hemagglutinin-neuraminidase (HN) proteins [25]. Both PAMPs stimulate a strong type I interferon response [26,27]. 

The consequence is (i) immunogenic rather than tolerogenic T-APC interaction and (ii) the induction of immunogenic cell death (ICD) [28]. The ICD involves immunogenic apoptosis, necroptosis, and pyroptosis. Pyroptosis is an inflammatory form of cell death that is triggered by inflammasomes [29]. The details of these complex processes of NDV-mediated viral oncolysis have recently been described [30].

## 4. Three Examples for Vaccination-induced Long-term Protective Anti-cancer Immunity

Evidence for induction of long-term protective immunity against cancer should include (i) the quantification of the effect in terms of % survival or median overall survival, (ii) comparison of the effect to a control group, and (iii) immunological basis of the effect.

Figure 1 shows three examples. Figure 1A illustrates the pre-clinical results obtained with the murine ESb metastatic lymphoma. Ten days after intradermal transplantation of the ESb tumor cell line into syngeneic DBA/2 mice, a small palpable tumor occurred, which was operated. Two groups of such mice were then vaccinated with irradiated ESb cells (control group) or irradiated NDV-infected ESb cells. While in the control group, all animals died within one month (no therapeutic effect, same curve as non-vaccinated mice), in the group that was vaccinated with virus-modified ESb cells more than 50% survived long-term [31]. This demonstrates the importance of virus modification of the tumor cell vaccine to obtain a therapeutic effect.

The results of Figure 1B were obtained with a human tumor cell vaccine similar to that of Figure 1A, meanwhile termed ATV-NDV, standing for the autologous live tumor cell vaccine modified by infection with NDV (non-lytic strain Ulster). The results were obtained from a randomized-controlled study of patients suffering from stage IV colon carcinoma with operated liver metastases, thus testing the immunotherapy as a tertiary prevention method. The vaccine was prepared from the cells of the operated liver metastases. The curves of % metastasis-free survival demonstrate that immunotherapy has a significant tertiary prevention effect [32,33,34].

Figure 1C shows recent results that were obtained from patients suffering from glioblastoma multiforme (GBM). The curves compare the median overall survival (median OS) of patients that were treated first line by standard therapy (radiochemotherapy according to the Stupp protocol, left arrow) to those treated first line by standard therapy in combination with multimodal immunotherapy, as performed at IOZK in Cologne, Germany (right arrow). The details have been described [35]. A new calculation was only done with patients that were comparable to the Stupp study. The difference in median OS (shown) was 8.8 months. The difference in % two-year OS (not shown) was 21,4 %. 

## 5. 50 Years of Clinical Application of NDV

A recent review [36] provides an overview of 50 years of basic and clinical research on oncolytic NDV with its particular anti-neoplastic and immune stimulating properties. The cancer patients were systemically treated as oncolytic virotherapy, or locally by NDV-based oncolysate vaccines, by live tumor cell vaccines (ATV-NDV) or by DC-based oncolysate vaccines (IO-VAC^R^). The clinical applications included single case observations, case series studies, and Phase I to II/III studies. The high safety profile of NDV is due to the lack of interaction with host cell DNA, independence of virus replication from cell proliferation, induction of immunogenic tumor cell death, and of a strong type I interferon response.

## 6. Mechanisms of TA Transport

### 6.1. TA Uptake and Transport via the Lymphatic System

The tumor vaccines are commonly applied to the skin, either intradermally or subcutaneously. This was the case also with the vaccines ATV-NDV and IO-VAC^R^. At these sites, resident immature DCs become activated by microbial products (e.g., NDV of the vaccine) to mature. Activated DCs in the skin (Langerhans cells) or dermis (dermal DCs) capture antigens (see Table 1). They then migrate through the epidermis and transport the antigen to regional lymph nodes. Thereby, the DCs mature and become efficient APCs. They turn down Fc- and mannose-receptors, whose principal function is antigen capture, and upregulate molecules involved in T cell activation such as CD80, CD86, ICAM-1 and IL-12. 

The likelihood for cognate interaction between an APC and the corresponding antigen-specific T cell is very low when considering the fact that the frequency of an antigen-specific T cell among the whole population of T cells is one in a million, if not lower. It is postulated that successful anti-tumor vaccination depends on cognate T-APC interactions. Multiple cognate interactions at different sites might augment the chance to reach such a goal. Therefore, more insight is provided here into the possible sites of cognate interaction: lymph nodes, spleen, and bone marrow (BM).

Naïve B and T lymphocytes from the blood enter lymph nodes via high endothelial venules (HEVs). Once arrived, they migrate to different areas following signals from chemokines that are produced in these areas and bind selectively to either cell type. This leads to the segregation of B cells into the B cell zone (lymphoid follicle) and T cells into the T cell zone (parafollicular cortex). The T cell-rich zones contain a network of specialized fibroblast cells, called fibroblast reticular cells (FRCs). Many of these form the outer layer of tubelike structures, called FRC conduits, 0.2–3 μm in diameter. These conduits serve to transport antigens from afferent lymphatics to T cell zones. Naïve T cells express the chemokine receptor CCR7. The corresponding chemokines CCL19 and CCL21 are produced by FRCs in the T cell zones of lymph nodes. DCs activated by microbes also express CCR7 and lymphatic endothelial cells express CCL21. This explains why DCs enter the node through lymphatics and migrate to the same area of the node as naïve T cells.

### 6.2. TA Uptake and Transport via the Blood to Spleen and Bone Marrow

The white pulp of the spleen has an anatomic arrangement of T cells, APCs, and B cells, which is optimized to promote the interactions that are required for adaptive immune responses, quite analogous to that described for lymph nodes.

New insights are especially relevant for understanding the unique properties of the BM. Its main function, hematopoiesis, is described in all textbooks of immunology. Less is described about certain important additional functions: (i) the initiation of primary T cell responses and (ii) the maintenance of memory T cells (MTCs) in specialized niches. 

There are estimates that approximately 12% of all lymphoid cells in the human body are found in the BM at any given time as compared to 2% in the peripheral blood [37]. Human adult BM with its red (medulla ossium rubra, hematopoietic) and yellow (medulla ossium flava, fat) marrow weighs about 2.6 Kg [38]. Approximately 8–20% of BM mononuclear cells are lymphocytes, with a T:B cell ratio of about 5:1. Within the BM stroma and parenchyma, the lymphocytes are diffusely distributed. They can also be condensed in follicles surrounding a blood vessel. In such follicles, DCs and T cells could be visualized by immunohistology [39].

LFA-1α and α4 integrins on T cells interact with VCAM-1, MadCAM-1, and ICAM-1 on BM stroma. Homing to BM also involves chemokines (e.g., CXCL12 (SDF-1α,β)) and cytokines. This is true for homing to BM of circulating T and B cells, DCs, and tumor cells [39,40,41]. The T cells transmigrate through endothelium via diapedesis into BM parenchyma.

BM contains resident DCs in its parenchyma that pick up blood-borne antigens, including TAs. Additionally, APCs can enter from the blood into the BM parenchyma. Cell-associated antigens are cross-presented by BM DCs much more efficiently than soluble antigen [42]. This is particularly true for virus-infected cells.

A hallmark of the immune system of vertebrates is its capacity to maintain a memory function for antigens that were once encountered. In a recent review, the lifestyle of memory plasma cells of the BM served as a paradigm. The persistence of memory cells is dependent on distinct survival signals. These are provided by individual mesenchymal stromal cells. Thus, memory is not defined by intrinsic “half-lifes”, but by cytokine-secreting stromal cells [40].

## 7. Bone Marrow and its Importance for T Cell Mediated Immune Responses to Blood-borne Antigens

### 7.1. Effect of Transient Dietary Restriction

Collins N et al., in a recent paper from Cell [43], showed how the immune system adapts to transient dietary restriction (DR), which causes nutritional stress. In the context of DR, MTCs totally collapsed in secondary lymphoid organs. In contrast, in the BM, MTCs accumulated. The BM response was coordinated by glucocorticoids and involved a state that is associated with energy conservation. It involved complete remodeling of the BM compartment with increases in T cell homing factors and adipogenesis. During DR, adipocytes, as well as CXCR4-CXCL12 and sphingosine-1 phosphate (S1P) interactions with its receptor S1P_1_R, contributed to enhanced T cell accumulation in BM. Another recent finding is that human adipocytes from BM display distinct immune regulatory functions [44].

### 7.2. Antigen Specific Cognate T-DC Interactions in the BM

Here follows a short summary about experiments demonstrating that BM is a priming site for T-cell responses to blood-borne antigen [39].
Naïve T cells (CD62L^high,^ CD44^low^,CD69^neg^) expressing a transgenic ovalbumin (OVA)-specific TCR were shown to home to BM parenchyma upon the transfer to naïve (B6) wild-type or splenectomized alymphoplastic mutant (*Map3k14*^aly/aly^) C57BL/6 mice. Upon OVA challenge (0,45 mg/mouse i.v.), BM DCs took up the blood-borne antigen and processed it via MHC I and II pathways. The transferred CD4+ or CD8+ T cells formed multicellular clusters with the BM-resident DCs (CD11c+) as APCs, became activated, proliferated, and differentiated into effector T cells and MTCs.BM responses could be generated in mice without lymph nodes and spleen. Thus, BM is autonomous in generating primary CD4+ and CD8+ T cell responses.In the absence of administered adjuvant, the BM responses were not tolerogenic and they resulted in generation of CTL activity, protective anti-tumor immunity, and immunological memory.

### 7.3. Antigen Processing and Presentation, Scanning of APCs by T Cells, Synapse Formation, APC-T Cluster Formation, CD4-CD8 T-T Interactions 

Intra-cellular events: Cytosolic proteins are degraded in the proteasome and the generated peptides delivered via transporter protein (TAP) to the endoplasmic reticulum (ER). Stable complexes of class I MHC molecules with bound peptides move out of the ER, through the Golgi complex, to the cell surface. Viral genes and mutated tumor genes can also generate peptides from within the cell. In this case, peptide channeling contributes to the high efficiency of class I immunosurveillance of tumors and intracellular pathogens. The translation of pre-spliced RNAs in the nuclear compartment via ribosomes generates peptides for the class I pathway [45]. In APCs, peptides from minute amounts of proteins are capable of outcompeting an excess of constitutively generated peptides [46]. Antigenic peptides are derived from a nuclear non-canonical translation event and are as efficiently produced from introns as from exons. They are independently regulated from the synthesis of full-length proteins [47]. The immunopeptidome is highly skewed from the cellular degradome. CD8+ T cells can recognize class I-associated peptides on all nucleated cells of the body.

The endocytic compartment is highly efficient in the processing and presentation of peptides via the MHC class II pathway. Here, proteins are proteolytically cleaved by enzymes in lysosomes and late endosomes. Newly synthesized MHC class II molecules are transported from the ER to the endosomal vesicles. When MHC class I- and II-restricted peptides are offered within the same carrier protein context, the endocytic compartment favors the presentation by class II by at least 1000-fold [48].

APC-T cell interactions: Upon APC contact, the mobile T cells scan the APC’s cell surface for the presence of exactly fitting pMHC complexes. This scanning process, which allows for the T cell to distinguish MHC bound non-self from self peptides, might obliviate the need to purify tumor-derived neopeptides. It also explains the low detection limit for T cell triggering [49]. Four pMHC per TCR cluster are sufficient for triggering. The vast majority of the about 10,000 presented peptides of an APC in vivo are normal self peptides.

Synapse formation: Once a cognate T-APC interaction event has taken place, an immunological synapse [50,51] is formed and the two types of cells stay together and exchange signals that are important for MTC formation. IFN-γ and TNFα are transmitted from the T cell to the APC and IFNα and IL-12 from the APC to the T cell. Bidirectional cell stimulation, survival, and antitumor activity have been described after cognate interactions between MTCs and TA-presenting DCs from BM of breast cancer patients [52]. 

APC-T cluster formation: APC scanning in the BM and bidirectional cell stimulation is followed by APC-T cluster formation, the generation of T lymphoblasts, to clonal T cell expansion within such clusters [39,52,53], and to the release of activated T cells from the clusters. Within 10 days, T cell mediated immune responses from the BM leads to the development of antigen-specific CTL activity and MTCs.

Similar is the sequence of events with CD4 helper T cell responses. The activation of transgenic CD4 T cells specific for human C reactive protein (hCRP) could be visualized in clusters in BM in situ upon cognate interactions with BM-DCs [54].

CD4-CD8 T-T interactions: Multicellular T-APC clusters have also been observed in lymph nodes [53], in the liver [55], and in tumor tissue [56]. Such clusters facilitate CD4-CD8 T-T cell interactions. CD4 T cell help is required for CD8 T cell memory and it involves CD25 [57,58] and CD40 [59] mediated costimulatory signaling. The ligation of CD40 on APCs via CD40L of Th1 cells greatly increases the APC’s co-stimulatory and CTL stimulatory capacity [60,61]. Thus, optimal priming of a CTL response with a “licence to kill” involves coordinated interactions of APCs with CD4 and CD8 T cells.

### 7.4. Therapeutic Potential of BM derived MTCs

The potential of re-activated MTCs from the BM of cancer patients and of T-APC interactions was tested in NOD/SCID mouse-human tumor xenograft systems [62]. BM derived naïve T cells (CD45RA+) or MTCs (CD45RA-CD45RO+) were separated and stimulated with autologous APCs to test which subset of T cells functions in vivo. These DCs were pulsed with breast tumor lysate. After 48 h, the co-cultured cells were transferred into NOD/SCID mice bearing autologous breast tumor and normal skin transplants. Tumor, but not skin tissue, became infiltrated by autologous MTCs, but not by naïve T cells. Many of the tumor-infiltrating MTCs expressed P-selectin glycoprotein ligand 1 (PSGL1). They were found around P-selectin+ tumor endothelium. Many of them also produced perforin. Additionally, clusters were seen in infiltrated tumors between MTCs and DCs. The human MTCs and the DCs, functioning as APCs, both expressed the chemokine receptor CCR7 in the tumor. The results demonstrated complete human tumor regression [56,62].

CTLs deliver a lethal hit signal towards target tumor cells at the cytotoxic secretory synapse via unidirectional perforin pore delivery [63]. CTLs kill multiple targets via the exocytosis of granzyme B containing cytotoxic granules, which are endocytosed and recycled in target cells [64]. This explains the high specificity and effectivity of CTL mediated anti-tumor responses.

What remains after the effector phase are MTCs whose subsets, migration patterns, and tissue residence have been described [65].

### 7.5. TA-specific Treg Cells from BM Exert Peripheral Tumor Immune Suppression

In addition to inducing antitumor effector T cell responses, the BM also induces TA-specific regulatory (Treg) T cells. These orchestrate peripheral effects, as exemplified in breast cancer. BM induced TA-specific Treg cells egressed from the BM via activation-induced peripheral homing receptor CCR2 and followed its ligand CCL2 that was secreted from breast cancer tissue [66]. 

## 8. Mechanisms of Maintenance of Long-term T cell Memory

### 8.1. Dynamics and Longevity of Memory

The bacterial lacZ gene product ß-galactosidase (Gal) served as a surrogate TA and the mouse ear pinna as a site of tumor resistance and optimal immunization potential [67]. The first T cell response that was observed after intra-ear pinna (i.e.,) inoculation of live lacZ transfected ESb (ESblacZ) tumor cells into syngeneic DBA/2 mice was observed in the BM. This response showed a peak after 10 days. It was analyzed by dominant Gal pMHC I tetramer analysis [39].

The ins and outs of MTCs from the BM were studied in this model system. Following T-cell priming (1. antigen contact response) in the BM, Gal-specific MTCs could be recruited from the BM to the peritoneal cavity by i.p. challenge with irradiated ESblacZ cells (2. antigen contact response). The secondary Gal-specific T cell response that is involved >80-fold enrichment of epitope-specific CD8+ T cells and the release of various cytokines [68]. Two months later, in these mice, the MTCs had returned from an activated state (mostly effector (E) MTCs) into a resting state (mostly central (C) MTCs) and from location in the peritoneal cavity back to the BM [69]. 

Gal-reactive peritoneal E MTCs, induced and re-activated as described above, were transferred from immunocompetent DBA/2 mice (primary host) to athymic nude (nu/nu) mice (secondary host) together with i.p. challenge of live ESblacZ cells (3. antigen contact response) to investigate the longevity of BM MTCs. The E MTC transfer again prevented tumor outgrowth and resulted in the long-term persistence of Gal-specific T cells in the BM and spleen. This process was repeated up to the 6. antigen contact in quaternary hosts, starting from DBA/2 derived E MTCs. While naïve nude mice died within 10 days following the injection of 1 × 10^5^ tumor cells, MTC-transfered nude mice were able to reject a tumor dose of 5 × 10^7^ and they survived longer than eight months [69]. 

### 8.2. Bone Marrow Niches for Maintenance of Memory T cells

The BM contains so-called “niches” that are made up of stromal cells. BM niches that sustain and modulate hematopoietic stem cells (HSCs) have known for a long time. More recently, it was discovered that BM also contains special microenvironmental domains or functional compartments (i.e., niches) for MTCs. BM MTCs express the key survival receptors IL-7Rα and IL-15Rβ. BM stromal niches provide the corresponding cytokines IL-7 [70] and IL-15 [71] for MTC survival.

Here follows a short update of recent findings concerning the BM and its function in immunological memory.
CD4+ MTCs helping antibody producing B cells were studied in the BM. In a secondary immune response to systemic antigen, antigen-specific helper T cells of the BM were found to aggregate together with MHC class II-expressing B cells. After 10 days, the immune clusters disappeared again. 30 days later, the expanded CD4+ MTCs returned to their BM niches and they were maintained there as resting cells [72].CXCR4 was found to be crucial for the entry of CD8+ T cells into the BM. This chemokine receptor also controls subsequent CD8+ T cell localization via attraction by CXCL12 (SDF-1α/β) toward BM niches, which support their survival [73].A hypothesis, recently proposed, suggests the existence of two niches in the BM to explain life-long T cell memory, one for T cell cycling and the other for T cell quiescence [74].A deuterium labelling study in mice supports a dynamic model for the maintenance of MTCs in the BM. This provides support for specialized BM niches. These are organized in such a way that MTCs can continuously self-renew and recirculate between the blood, BM, spleen, and lymph nodes [75].

### 8.3. Tissue-resident Memory T cells (T MTCs)

Understanding long-term protective anti-tumor immunity to be induced by cancer vaccines requires knowledge regarding MTCs, their subsets, tissue-distribution, and maintenance. Tissue-resident MTCs (T MTCs) are a recently described new subset of MTCs. 

They were discovered, among others, in murine and human BM and are polyfunctional cytokine producers. These cells, being dependent on IL-15, reside in BM parenchyma (“chilling in the bone” [76]). They represent a pool of resident MTCs in close contact with the blood circulation and expandable upon peripheral or systemic antigen re-challenge [77]. 

T MTCs specific profile of transcription factors include Runx3, Notch, Hobit, Blimp1, BATF, and AHR [78]. In the periphery, T MTCs are defined by expression of CD103 (α_E_β7 integrin), CD49a (VLA-1 or α_1_β_1_ integrin), and C-type lectin CD69. The retention of these cells in non-lymphoid tissues and solid tumors likely depends on the expression of these molecules. CD103 binds to epithelial cell E-cadherin. This interaction is required for polarized exocytosis of lytic granules. Natural or cancer-vaccine induced T MTCs directly controlled tumor growth in tumor models [78]. 

Skin CD8+ T MTCs have recently been reported to amplify anti-tumor immunity by triggering antigen-spreading through DCs. Antigen-specific activation of skin T MTCs led to the maturation of dermal DCs and their migration to draining lymph nodes for antigen cross-presentation [79]. 

Lung-resident protective CD8+ T MTCs could be induced with a nanoparticle vaccine. This was pH responsive and it could deliver at the same time and location a protein antigen and a nucleic acid adjuvant. Its application enhanced the magnitude and longevity of the specific T MTC response in the lungs [80].

Human small intestine intraepithelial and lamina propria CD8+ T MTCs have recently been described to persist for years. These cells were potent cytokine producers and they efficiently expressed cytotoxic mediators after stimulation [81]. 

### 8.4. Stem Cell-like Memory T Cells (S MTCs)

Another type of MTC is the stem cell-like MTC (S MTC). A substantial proportion of BM MTCs are S MTCs that are characterized by CD69 and CD127 expression and efflux capacity [82]. BM-resident S MTCs exhibit much higher levels of antitumor activity than spleen-resident respective cells [83]. While the maintenance of CD8+ MTCs in the spleen is dependent on cell proliferation, their maintenance in BM is independent from cell proliferation [84].

Human memory CD8+ T cell effector potential has been recently described to be epigenetically preserved during in vivo homeostasis. Whole-genome bisulfite sequencing of primary naïve, short-lived E MTC, and longer-lived C MTC and of S MTC CD8+ T cells identified effector molecule genes with demethylated promoters and poised for expression. Effector-loci DNA demethylation was heritable and preserved during IL-7 and IL-15 mediated in vitro cell proliferation. Conversely, cytokine-driven proliferation of C MTCs and S MTCs resulted in phenotypic conversion into E MTCs and it was coupled with increased methylation of CCR7 and Tcf7 loci [85].

In contrast to activated T cells, which, after adoptive T cell therapy, may rapidly become tolerant, due to anergy, senescence, and/or exhaustion [86], S MTCs cells are strongly resistant to tolerance. Recently, a simple in vitro co-culture procedure has been described to convert activated T cells into S MTC cells. The stimulating cells were OP9 stroma cells expressing Notch ligand [87]. Another study used IL-7 and IL-15 to instruct the generation of human S MTCs from naïve precursors in vitro [88].

## 9. Recruitment and Re-activation of MTCs from BM by Virus-modified Tumor Vaccine

Primary operated breast cancer patients contain, in their BM, cancer-reactive MTCs that can be re-activated ex vivo by DC-based APCs and exert therapeutic potential in human tumor xenotransplant models [56,62]. Further studies revealed that DCs that were pulsed with viral oncolysates stimulated significantly higher MTC ELISPOT responses than DCs that were pulsed with tumor lysate without virus (NDV) infection. The supernatants of co-cultures with viral oncolysate-pulsed DCs contained augmented titers of IL-15 and IFN-α [89].

Furthermore, previous animal studies had revealed that ATV-NDV vaccine stimulated significantly higher CTL responses from MTCs than ATV vaccine without virus (NDV) infection [90]. The increased response was due to secreted IFN-α [91]. Apart from type I IFN, also cytokines released during a secondary immune response, such as IL-2, IL-12, IL-15, IL-18, and IL-21, determine the memory potential of antigen-specific CD8+ T cells [92]. The CD8+ T cell response to a solid tumor required the availability of a third signal provided by IL-12 or type I IFN and their corresponding receptors [93].

IL-2, IL-7, and IL-15, which share the common gamma chain cytokine receptor, shape the T cell response to cognate antigen and the ensuing maintenance of MTCs [92]. IL-7 and IL-15 are important for the homeostatic survival of CD4+ and CD8+ MTCs [88]. They also produced enhanced anti-tumor activity of CAR.CD19 T cells and increased their resistance to cell death [94]. As already mentioned, IL-7 and IL-15 instruct the generation of human S-MTCs from naïve precursors [88].

The dynamics of recruitment of MTCs from the BM, of their re-activation in the periphery, and of their return into a resting state and into the BM has been extensively studied in an animal model [69].

The results of a prospective randomized trial demonstrated the efficiency of post-operative adjuvant vaccination with ATV-NDV in colon cancer patients following the resection of liver metastases [32]. The mechanism of function of long-term patient survival has been discussed [33].

## 10. Discussion

Of importance for tumor vaccine design is the question of polarization of the response to be induced. A difference exists between the immune responses against extra-cellular (Th2, B cells, antibodies) versus intra-cellular (Th1, CTL) microbes. As TAs as pMHC complexes are of intra-cellular origin, their cognate immune response should be guided by TA presentation via DCs that are polarized towards Th1 and CTL responses rather than by B cells and antibodies.

Another question is the choice of a relevant cancer target in a tumor vaccine. Should it be a purified individual neoantigen or would it be sufficient for providing an autologous tumor lysate for processing and presentation by DCs? It has been documented that migratory TA-specific T cells, by scanning an APC for optimal key-lock fit, can distinguish TA-derived pMHC complexes from self-protein-derived pMHC complexes. Two-photon intravital microscopy revealed that CTLs only infiltrate solid tumors in depth when the tumor cells express the cognate antigen [95]. APC scanning is a physiological process and explains why minute amounts of TA derived pMHC complexes are sufficient for eliciting a T cell-mediated response. The source of TAs should be autologous since neoantigen derived pMHC complexes are individually unique.

Tumor vaccine design also has to consider the question of directing the response against one or several TAs. One lesson from basic immunological studies is that one antigenic determinant is not sufficient for protein immunogenicity [6]. Two antigenic determinants, a hapten that is by a hapten-carrier presenting B cell and a pMHC class II restricted carrier determinant recognized by a Th2 cell, were the minimal requirement. With regard to a TA specific T cell response, the two antigenic determinants to be recognized by a CD8 CTL and a Th1 T helper cell could be derived from the same or from different TAs. Another point is that the cancer-reactive T cell memory repertoire has been found to be polyspecific and individually distinct [96]. To re-activate such a repertoire would require stimulation by several individually selected TAs or by autologous DCs cross-presenting autologous tumor lysate or autologous viral oncolysate. 

Naïve T cell stimulation requires, apart from cognate antigen interaction, costimulatory signals. These can be provided from activated innate immune cells. Which strategy might be optimal for providing costimulatory signals? There is the choice between one or several costimulatory signals and the question of whether the inducing agent can be a live attenuated microbe or should be a purified subunit. Live attenuated microbes are often more effective in long-term protection against infectious diseases than subunit vaccines. Therefore, it can be postulated that live attenuated NDV as adjuvant in a cancer vaccine is superior to subunits, such as agonists to RIG-I [97] or Toll-like receptor (TLR). The molecular details of stimulating innate immunity via live attenuated NDV include (i) the activation of NK cells via HN interacting with NKp46 [98], (ii) activation of monocytes, macrophages, and DCs via NFκB, thus inducing a module for pro-inflammatory cytokines (TNFα, IL-2, IL-15, IFNα, IFN-γ) [99,100], (iii) re-programming DCs within 18 hrs of infection into polarization towards DC1 [101], (iv) upregulation of tumor necrosis factor-related apoptosis-inducing ligand (TRAIL) [102], and v) the induction of nitric oxide (NO) [99]. TCR- and costimulatory receptor-mediated signals are not additive, but they form a network that is branched, diversified, and bounded [103].

Thus, an attenuated strain of a bird paramyxovirus, oncolytic NDV, serves as a powerfull adjuvant in a therapeutic vaccine against cancer. Attenuated strains of human paramyxoviruses such as measles and mumps virus, although with immunosuppressive properties, are used successfully to eradicate these diseases by prophylactic vaccination. Paramyxoviruses appear to have properties well suited for vaccination purposes.

In addition to the response activation via CD4+ and CD8+ immune T-T cell interaction [90], there is the type I IFN response that influences the quality of the response. IFNα,β was reported to play an important role in the generation of a CTL response. The application of IFN neutralizing antibodies had a dramatic suppressive effect on the CTL response in vitro and also in vivo in mice [91]. CD8 T cells receive via binding of type I interferon to its cell surface receptor IFNAR1 an important signal 3 for survival [93] and the development of effector functions [104].

The survival curves of Figure 1A demonstrate the strong influence of NDV infection on the immunogenicity of the tumor vaccine. The modification of tumor cells by a low dose of NDV was reported to strongly augment a tumor-specific T cell response as a result of CD4+ and CD8+ immune T cell cooperation [90]. The survival curves of Figure 1B,C suggest that similar mechanisms might become activated in cancer patients that are treated by NDV modified tumor vaccine or by NDV oncolysate modified DC vaccine. 

The multimodal immunotherapy protocol that was developed at IOZK [23,35] involves the systemic application of oncolytic virus to induce ICD and local application of the vaccine IO-VAC. It is likely that this procedure leads to TA uptake and transport via the blood and the lymphatic system, thus increasing the chance of cognate T-APC interactions at different sites, such as lymph nodes, spleen, and BM. This might also increase the number and diversity of MTCs, such as central and effector subsets, as well as tissue-resident and stem cell-like MTCs.

Some classes of OVs may be more effective at the induction of anti-tumor immunity than others. However, no comparative studies exist so far. Most detailed information about the immunobiology of an OV seems to exist with regard to NDV. It has recently been extended and updated [105]. In addition to having broad immune stimulatory effects, NDV is the first described OV with the potential to break cancer therapy resistance. This includes resistance to chemotherapy or radiotherapy, resistance to apoptosis, resistance to hypoxia, and resistance to TRAIL. In addition, NDV was shown to be capable of breaking T-cell tolerance, resisting to anti-viral immunity, and resisting immune checkpoint blockade [30]. 

The results from clinical application of immune checkpoint inhibitory (ICI) antibodies has acquired significant attention in recent years at oncology conferences. A main reason has been the surprise that it was possible to obtain long-term survival benefits in late-stage diseases. However, the strategy of inhibiting negative signals to T cells interferes with physiological immune regulatory mechanisms and, therefore, provides increased risk to generate autoimmune diseases. Another problem is the non-targeted delivery. The side effects of this approach range from WHO grade 1 to 4, whereby 3 and 4 are very severe [106]. 

One message of this review is that post-operative vaccination with virus-modified autologous cancer vaccines can achieve long-term survival benefits. An important difference to ICI antibodies relates to the side effects. With the approach of vaccination, the observed side effects of WHO grade 0–2 are negligible and no auto-immune phenomena are observed [107]. 

A combination of approaches, such as 1. vaccination and/or oncolytic viruses to turn “cold” into “hot” tumors and 2. ICI might in future lead to further improvements. In support of this, RIG-I activation has been reported to be critical for responsiveness to checkpoint blockade [108].

A systematic review of specific immunotherapy studies in renal cancer identified 14 controlled studies involving 4013 patients. A meta-analysis of seven studies revealed that patients with specific immunotherapy had significantly higher OS than those in the control group. Active-specific immunotherapies involved five autologous tumor cell vaccines, one peptide-based vaccine, one virus (NDV)-based vaccine, and one DC-based vaccine [109].

## 11. Summary

This review updates information concerning the basics of long-term protective T cell mediated anti-tumor immune mechanisms. There exists a dichotomy of antigen-processing pathways in cells (cytoplasm and vesicles), of peptide-presenting MHC molecules (class I and class II), of peptides (nonamers and longer ones), and of TCR co-receptors (CD4 and CD8). This dichotomy provides the molecular basis for CD4-CD8 T-T cell interaction and the recognition of more than one antigenic determinant. Particular attention is given (i) to cognate interactions of TA-specific T cells with TA-laden DCs in T-APC clusters, (ii) to the role of microbe-derived T cell costimulatory signals, type I interferons, and cytokines, and (iii) on the mechanisms of long-term maintenance of tumor-reactive memory. The dynamics and longevity of memory T cell function appears to rest on tissue-resident and stem cell-like memory T cells. 

Chemokines and their receptors play an important role in directing cognate T-APC interactions in lymph nodes and spleen (e.g., CCR7 and CCL21/19), or in bone marrow (e.g., CXCR4 and CXCL12 (SDF-1α/β), or attracting Treg cells towards tumor tissue (e.g., CCR2 and CCL2 (MCP-1)). 

The review provides three examples of tumor immunotherapy-mediated enhancement of long-term survival. The selected studies employ peripheral immunization with NDV-modified vaccines with or without the systemic application of oncolytic NDV. 

Long-term survival rests on long-term immunological memory. The bone marrow protects and optimizes immunological memory. Its complex spongoid vascular system and parenchyma provide distinct niches for memory B and memory T cells. The homing of T cells to bone marrow involves integrins that interact with VCAM-1, MadCAM-1, and ICAM-1 on vascular endothelium. Bone marrow is autonomous in generating primary CD4+ and CD8+ T cell responses to blood-borne antigens and it provides niches that produce IL-7 and IL-15 for memory T cell survival and homeostasis. 

## 12. Conclusions

The induction of long-term protective immunity against tumors is based on double recognition of TAs by CD4+ and CD8+ T cells and on immunological memory. Innate and adaptive arms of the immune system have to act in concert. Bidirectional cell stimulation occurs upon cognate interactions between APCs and T cells in lymph nodes, spleen, and bone marrow. The latter organ contains niches that are specialized to maintain longevity of memory T and memory B cells. 

Immunotherapy of cancer has made tremendous progress in the last decade, as evidenced by Nobel Prizes in 2011 and 2018 and by the results from clinical studies. The main strategies are
(i)to release the tumor induced brakes on T cells by targeted checkpoint inhibitory antibodies,(ii)to boost instruction of the immune system via TA containing vaccines,(iii)to boost instruction by DC vaccines, thus bypassing the process of in situ antigen processing,(iv)to boost recognition bypassing instruction, the field of adoptive T-cell therapy, and(v)to boost recognition bypassing instruction and TA pMHC presentation, as exemplified by chimeric antigen receptor (CAR) T cells, bispecific T cell engagers (BITEs), or superantigens (SAGs).

Cancer vaccines that are modified by infection with a virus, such as NDV, boost the instruction of the immune system and are capable of inducing long-term protective anti-tumor immunity. The bone marrow and its special role in immunological memory play an important role in the maintenance of protective immunity.

Special attention should be awarded to the topic of side effects. The majority of new drugs approved in the last decade by the FDA can cause side effects as strong as WHO grade 3 and 4. In contrast, physiological approaches that boost instruction usually cause side effects of WHO grades <2.

Future improvements of cancer vaccine and oncolytic virotherapy can be expected from genetic engineering and from incorporation of additional therapeutic genes. This might improve specificity, potency, and delivery [110]. Improved specificity might derive from tumor neoantigens. Improved potency and delivery is expected from enhanced killing of tumor cells in metastatic cancer, from better intravenous transport, better tumor targeting, and better intratumoral virus dissemination.

## Figures and Tables

**Figure 1 biomedicines-08-00055-f001:**
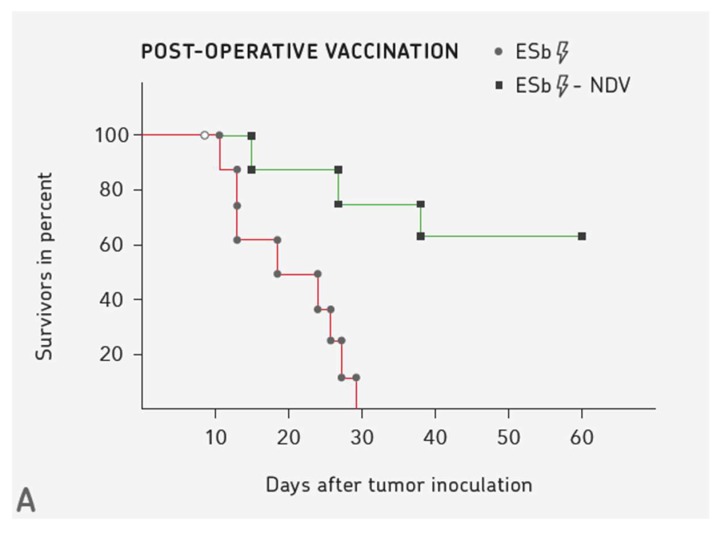
Examples of therapeutic benefits from immunotherapy with virus-modified cancer vaccines (green) in comparison to controls (red). (**A**) Comparison of a virus-modified to a non-modified tumor cell vaccine in a murine tumor model of a highly metastatic lymphoma cell line [31]. 10^5^ ESb lymphoma cells were transplanted intradermally into 24 syngeneic DBA/2 mice at day 0. Ten days later, the palpable tumor was removed. All the mice from a non-vaccinated group and also those from the group vaccinated with the non-modified tumor vaccine died within 1 months. A therapeutic effect was obtained only in the group vaccinated with ESb tumor cells that had been infected with the avian virus NDV. (**B**) Results from a randomized-controlled Phase II/III clinical study of stage IV colon cancer patients after resection of liver metastases [32,33]. The objective was to test the effect of post-operative vaccination as a tertiary prevention method. The virus-modified vaccine ATV-NDV was similar to that of A. There was a significant benefit from post-operative vaccination in overall survival and in metastasis-free survival, as evaluated after a ten-year follow-up period. (**C**) Comparison of first-line post-operative treatment of patients suffering from glioblastoma multiforme (GBM) by radiochemotherapy versus radiochemotherapy plus immunotherapy. The immunotherapy performed at IOZK was multimodal as described [23] and included systemic NDV application in combination with moderate electro-hyperthermia (mEHT) to induce ICD and vaccination with a dendritic cell vaccine containing autologous TAs and NDV (IO-VAC). The retrospective analysis of comparable patients was kindly performed in Nov 2019 by Dr. Stefaan van Gool. The curve shows median overall survival (OS) of 23.4 months (right arrow) from GBM patients (*n* = 34) treated by radiochemotherapy plus multimodal immunotherapy. The left arrow points to median OS of 14.6 months obtained by radiochemotherapy alone according to the Stupp protocol with temozolomide. To compare median OS to percent overall survival, the results of OS at two years were: 47.9% with immunotherapy versus 26.5% without immunotherapy.

**Table 1 biomedicines-08-00055-t001:** Human Subsets of Dendritic cells and their Functions.

Feature	pDC	cDC	iDC
Surface CD	CD123	CD11c	CD11c
TF	TCF4	IRF4	IRF8
Cytokine	IFNα,β	IL-12	IL-2
APC function for	viruses	bacteria	virus-infected cells
Routes	infection	extra- or intracellular ^1^	cross-presentation ^2^
Loaded MHC	class I	class II	class I
Cognate T cell	CD8	CD4	CD8

Major characteristics (knowledge still incomplete). pDC plasmacytoid DC; cDC conventional DC, iDC inflammatory DC; CD cluster of differentiation marker; TF transcription factor; IRF interferon regulatory factor; Induction of IRF8 in the common DC progenitor is required for type 1 (DC1) fate specification [13]; ^1^ uptake into the endocytic system via receptor-mediated phagocytosis or via macropinocytosis; ^2^ uptake by ingestion, transport into cytosol, processing in proteasomes and peptides presented by class I MHC together with upregulated costimulator to CD8+ T cell; antigens from virus-infected cells can also enter endosome/lysosome vesicles via macropinocytosis to produce peptides that are presented by class II MHC to CD4+ T cells.

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
