# Peer review of "New Insights into Mechanisms of Long-term Protective Anti-tumor Immunity Induced by Cancer Vaccines Modified by Virus Infection"

_biomedicines, 2020, doi:10.3390/biomedicines8030055_

Round 1
Reviewer 1 Report
In an era where cancer immunotherapy has given new hope for curative treatment for aggressive malignancies, the elucidation of the mechanisms involved in induction of long-term systemic immune responses can bring new insights towards directing the new generations of treatment. This review gives a deep dive into the many aspects of tumor immunity, with an emphasis on utilizing viruses as cancer vaccines for long-term protection. This is timely and relevant to the field. One shortcoming of this work, is that the review focuses mainly on the author’s own work using NDV as the basis of a cancer vaccine, with very little reference to other virus-based cancer vaccines that are under investigation. Therefore, the title and abstract are somewhat deceptive, since they lead the reader to believe that the manuscript is a more general review on virus-based cancer vaccines. The review would be interesting for a much broader audience if it would broaden its scope to include other viruses and perhaps even discuss whether some classes of viruses could be more effective at induction of anti-tumor immunity than others. Alternatively, the title and abstract should at least be altered to more accurately reflect the content of the review.
A further point is that, while the paper nicely begins and ends with a focus on how virus infection can be utilized as tumor vaccines, the middle part gets quite deep into the immunology, and we lose the context of how these different aspects (for example, bone marrow-derived versus tissue resident memory T cells) relate back to the point, which was virus infection as a mediator of tumor vaccination. I would suggest trying to tie these sections back into main point and possibly eliminating some aspects that are too technical and not really relating to the tumor vaccination.
Author Response
- At the end of the Introduction (line 44-49) the author explains that cancer vaccines involving genetic engineering are not the topic of this review.
- Whether some classes of Virus could be more effective than others is discussed (line 531-538).
- A new Paragraph (9.) makes a link between the virus-modified vaccine and the aspect of recruitment and activation of MTCs, including those from the BM.
Reviewer 2 Report
The author has put together a vast amount of literature on anti-tumor immunity which can be enhanced by cancer vaccines modified by virus infections such as Newcastle disease virus. The author has also described how the bone marrow is a special niche for initiation, propagation and maintenance of T cell memory, the most important arm of anti-tumor immunity. The review is very well written and only minor changes are necessary.
- The author can describe the literature search, inclusion criteria etc for the review.
- Line 25: Like the brain with “its”
- Line 26: prerequisites for “effective vaccination and long-term protective immunity”
- Line 31: in the molecular recognition of antigens by T cells: an antigenic fragment (peptide) which forms a complex …
- Line 79 : molecules distinguish the mode …
- Line 92: DCs(iDCs) can ingest virus-infected cells ..
- Line 95: vaccine whose administration induces long-term protective anti-tumor immunity, these vaccines will be described in the following section.
- Line 96: Typo is it iDC or DC1?
- Line 99: Interaction of these cytokines
- Line 105: I think the gene name for CD137 can be omitted
- Line 106 : regulators of metabolism.
- Line 116: at IOZK, Cologne
- Line 124-Line 125 is difficult to understand, kindly edit this point.
- Line 131: The ICD involves immunogenic …
- Line 176: thus testing the immunotherapy
- Line 191: become activated by microbial products (e.g. NDV in the vaccine).
- Line 234 – Line 242 can be moved after Line 228. The paragraph Line 229 to Line 233 can be moved down.
- It would be of great interest to readers how bone marrow provides distinct survival signals for persistence of memory lymphocytes, the author can kindly add a point or two on this.
- Line 246: Collins N et al (Cell xxxx) showed how …
- Line 389 – Line 396: Could the author kindly include the molecular mechanism behind how the T MTCs (which do not recirculate) were detected in the bone marrow?
- Line 416: S MTCs which are characterized …
- Line 549-Line 551 and conclusion seems to be end abruptly, could be modified.
Author Response
All suggestions (thank`s) are incorporated.
Point 18. How BM provides distinct survival signals (e.g. IL-7, IL-15) for MTCs and more points are added in a new Paragraph (9.): Line 442-467
Point 20. The different subsets of MTCs were separated first by a cell sorter and then analyzed.
Line 603-607: A Paragraph concerning future improvements has been added to the Conclusions.
Reviewer 3 Report
The present manuscript by Volker Schirrmacher entitled “New insights into mechanisms of long-term protective anti-tumor immunity induced by cancer vaccines modified by virus infection” describes the mechanism involve behind the long-term protection induced by two tumor vaccines modified by virus infection (ATV-NDV and IO-VAC) highlighting studies performed in animal models and humans.
Minor improvements needed:
The author describes in detail the basic mechanisms involved in the immune response. I would suggest including references for well-established immunology concepts.
The author showed interesting data on the benefits of immunotherapy with virus-modified cancer vaccines. It would be of interest if the author shows data proving the mechanisms involved in the long-term protection i.e. longevity of memory T cells after vaccination with virus-modified cancer vaccines.
Given the expertise of the author in the field of virus-modified cancer vaccines, I will suggest mentioning what needs to be improved from these vaccines and what studies need to be done.
Author Response
- References for well-established immunology concepts are included: 1, 2, 7, 8, 14,15, 20, 50, 60, 65, 86, 93
2. A new Paragraph (9.) summarizes points linking long-term protection with longevity of MTCs and re-activation by virus-modified cancer vaccine (lines 442-467)
3. A paragraph concerning future improvements has been added to the Conclusion. (lines 603-607).